# DYNAMIC REPRESENTATION OF OPTIMAL TRANSPORT VIA ENSEMBLE SYSTEMS

## ABSTRACT

Optimal transport has gained widespread recognition in diverse areas from economics and fluid mechanics, lately, to machine learning. However, its connection and potential applications to the domain of dynamical systems and control remain underexplored. To fill this gap, we establish an ensemble-systems interpretation for modeling the optimal transport process. We interpret displacement interpolation of the transport between continuous distributions as a dynamic process and show that this can be modeled as an ensemble control system. This is achieved by establishing moment kernel representations for describing the dynamics of optimal transport and ensemble systems. This methodology further gives rise to an optimal transport based algorithm for learning controls for ensemble systems.

## 1 INTRODUCTION

Optimal transport (OT) has gained great popularity owing to its broad applicability to diverse scientific domains, ranging from economics to fluid mechanics (Galichon, 2016; Hassanzadeh et al., 2014). In recent years, interest in utilizing OT techniques in the domain of machine learning has seen a particularly stellar growth, and research and development remains persistent (Villani, 2009; Seguy et al., 2017; Torres et al., 2021). To promote new thoughts on the interplay between OT and machine learning, this work is devoted to bridging the ideas of dynamical systems and control to OT.

Although transport between distributions or states of physical or engineered systems is a canonical task in dynamical systems and control, the connection of these fields with OT remains underexplored with sparse literature. Existing studies focused on the use of OT formulations to tackle problems involving optimal control, state estimation, linear-quadratic-Gaussian control, and multi-agent systems (Taghvaei & Mehta, 2016; Chen et al., 2016; Haasler et al., 2021). However, interpretation and modeling of OT by using dynamical control systems remains an open problem.

Driven by the desire to fill this literature gap for stressing the role of dynamical systems and control in OT and OT-based machine learning, we propose to build a machine learning model to represent OT in terms of dynamical systems. In turn, this model also enables and facilitates learning for control systems through the lens of OT. The integration of OT, machine learning, and dynamical systems proposed in this work provides new insights and tools to advance these interdisciplinary fields.

**Contributions.** (1) Based on the time-dependent description of OT in terms of displacement interpolation (DI), we construct a machine learning model that represents an OT process in the form of an ensemble control system defined on a function space. In this model, the control inputs are learnable parameters used to track the OT dynamics; (2) To effectively train the model, we develop the moment representation of the model, which draws a parallel between model training and optimal control of the moment-parameterized system; (3) Consequently, the task of representing OT is mapped to learning an optimal control that drives the ensemble system along the DI trajectory connecting the source and target distributions.

The paper is organized as follows. In Section 2, we briefly review OT and introduce DI from the perspective of dynamical systems. In Section 3, we construct the desired machine learning model for representing OT in terms of an ensemble system, and then introduce the moment parameterization

to train the model by learning an optimal control for the ensemble system. Section 4 is dedicated to simulation examples for demonstrating the applicability of the proposed framework.

## 2 TIME-DEPENDENT OPTIMAL TRANSPORT

OT is concerned with transporting one probability distribution to another with the minimal cost. Mathematically, in Monge's formulation, this can be formulated as a constrained optimization problem, minimizing the cost $c$ subject to the desired transport from a probability measure $\mu$ on $X$ to another probability measure $\nu$ on $Y$, given by

$$
J_{OT} = \min_{\Phi:X \to Y} \int_X c(x, \Phi(x))d\mu(x),
$$
$$
\text{s.t.} \quad \nu = \Phi_\sharp \mu, \tag{1}
$$

where $X$ and $Y$ are Polish spaces, i.e., completely metrizable separable topological spaces, $c : X \times Y \to \mathbb{R}$ is lower semicontinuous, and $\Phi_{\#}\mu$ denotes the pushforward of $\mu$ by the map $\Phi : X \to Y$, and thus a measure on $Y$ satisfying $(\Phi_{\#}\mu)(B) = \mu(\Phi^{-1}(B))$ for any measurable set $B \subseteq Y$.

This transportation process can be interpreted in a dynamic fashion through the ideas of DI. To elaborate, it is known that the transport cost depends on the path connecting the source and target measures, namely,

$$
J_{DI} = \min_{\{\Phi_t:X \to X\}_{0 \le t \le 1}} \int_X C(\Phi_t(x))d\mu(x),
$$
$$
\text{s.t.} \quad \Phi_0 = I, \quad (\Phi_1)_{\#}\mu = \nu, \tag{2}
$$

where $I$ is the identity map on $X$, and $\Phi_t(x_0)$ defines a curve in $X$ starting from $x_0 = \Phi_0(x_0)$ at $t = 0$ with $x_1 = \Phi_1(x_0)$ at $t = 1$ (Peyré & Cuturi, 2017). To enforce the equivalence between (2) and (1), it suffices to choose $C(\gamma_t) = \int_0^1 c(\dot{\gamma}_t)dt$ for $\gamma(t) \in X$. In addition, if $c$ is a strictly convex function on $\mathbb{R}^n$, then it satisfies $c(y-x) = \inf_{\gamma(t)}\{\int_0^1 c(\dot{\gamma}_t)dt : \gamma_0 = x, \gamma_1 = y\}$, and the infimum is achieved uniquely by the straight line, i.e., $\gamma(t) = (1-t)x+ty = (1-t)\Phi_0(x)+t\Phi_1(x)$ connecting $x$ and $y$. Consequently, this gives the time-dependent transport function $\Phi_t = (1 - t)I + t\Phi_1$, and the DI $\rho_t = \left[(1 - t)I + t\Phi_1\right]_{\#}\mu$ between $\mu$ and $\nu$ (Villani, 2021).

As a simple illustration, if $X = Y = \mathbb{R}$ and $c(x,y) = |x - y|^2$, then the transport map can be explicitly calculated, given by $\Phi = G^{-1} \circ F$, provided continuity of $F$, where $F$ and $G$ are the cumulative distribution functions of $\mu$ and $\nu$, respective, and $F^{-1}$ and $G^{-1}$ are the respective *generalized inverse*, defined by $F^{-1}(s) = \inf\{x \in \mathbb{R} : F(x) > s\}$ and $G^{-1}(s) = \inf\{x \in \mathbb{R} : G(x) > s\}$ (Thorpe, 2019). Then, the DI between $\mu$ and $\nu$ is given by $\rho_t = \left[(1-t)I+tG^{-1}\circ F\right]_{\#}\mu$ with $\rho_0 = \mu$ and $\rho_1 = \nu$. In this case, the OT cost is $J_{DI} = J_{OT} = \int_0^1 c(F^{-1}(s), G^{-1}(s))ds$, which coincides with the Wasserstein distance $W(\mu, \nu)$ between $\mu$ and $\nu$.

## 3 ENSEMBLE-SYSTEMS INTERPRETATION AND MODELING OF OPTIMAL TRANSPORT

The time-dependent description of OT presented in Section 2 reveals its intimacy with dynamical systems. Specifically, the map $\Phi_t$ in (2) represents the flow of a dynamical system evolving on $X$ with the specified endpoints and $\rho_t$ is the trajectory (geodesic) realizing the dynamic process of OT. Although OT and its dynamic descriptions have been extensively investigated (Chen et al., 2021; Montesuma et al., 2023), an unexplored essential question concerning how to build a dynamical systems representation that models the time-dependent OT process, and, on the other hand, leverage the notion and tools of OT to enable learning controls for steering dynamical systems remains unanswered.

We will address this fundamental two-way question. The solution will fully empower the capacity of OT by integrating it with theoretical and learning-based dynamical systems tools, e.g., realization theory, reservoir computing, recurrent neural networks, and neural ordinary differential equations (Miao et al., 2022; Chen et al., 2018; Chang et al., 2018). Moreover, it will also expand the scope

of OT to tackle problems involving analysis and control of dynamical systems, e.g., reinforcement learning and optimal control design (Narayanan et al., 2019; Yu et al., 2020), which were remote applications of OT. To this end, in this paper, we propose an ensemble systems interpretation that enables a unified modeling framework for representing dynamic OT processes.

### 3.1 A GENERALIZED OPTIMAL TRANSPORT LEARNING MODEL

Because OT can be considered as a dynamic process concerning with transporting one probability distribution to another over a probability space along the associated DI trajectory, it is meaningful to represent this process by using a dynamical system defined on a function space. Here, we formulate a *generalized OT-learning model* (GOTLM), that is, a parameterized linear control system described by a family of ordinary differential equations of the form,

$$\frac{d}{dt}x(t,\beta) = A(\beta)x(t,\beta) + \sum_{i=1}^{p} b_i(\beta)u_i(t). \tag{3}$$

We refer this to as an *ensemble system*, where $A(\beta) \in \mathbb{R}^{n \times n}$, $b_i(\beta) \in \mathbb{R}^n$, and $\beta \in \Omega \subseteq \mathbb{R}$ denotes the system parameter, $x(t, \cdot) \in \mathcal{F}(\Omega, \mathbb{R}^n)$ is the state of the system defined on the state-space $\mathcal{F}$, and $u_i(t) \in \mathbb{R}$, $i = 1, \ldots, p$, are the control inputs. Here, in our exposition, it is sufficient to consider the ensemble system defined on $\mathcal{F} = L^p(\Omega, \mathbb{R}^n)$. A typical goal of controlling an ensemble system as in (3) is to learn the $\beta$-independent controls $u_i(t)$ that transport the ensemble from an initial functional form (configuration), $x_0 \in \mathcal{F}(\Omega, \mathbb{R}^n)$, to a target configuration, $x_T \in \mathcal{F}(\Omega, \mathbb{R}^n)$, at some time $T$ (Li & Khaneja, 2009; Li, 2011; Kuritz et al., 2019; Chen, 2019).

This learning representation model maps the input $x_0$ to the output $x_T$ tracking the OT trajectory by tuning the control inputs. Specifically, the functions $A$ and $b_i$ are hyperparameters, and the controls $u_i(t)$ are learnable parameters used to train the model. In the following, we will establish a systems-theoretic approach to train GOTLMs as in (3) which represent the time-dependent transport from an initial probability density function (model input) to a desired target probability density function (model output). The mechanism of the proposed GOTLM is illustrated in Figure 1.

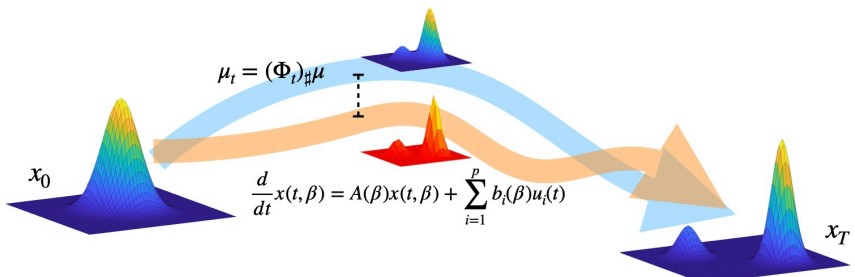

Figure 1: Mechanism of GOTLM.

### 3.2 MOMENT REPRESENTATIONS OF THE GENERALIZED OPTIMAL TRANSPORT LEARNING MODEL

Finding an appropriate parameterization (or representation) of a learning model is an essential step to facilitate learning and data-analytics tasks. Because the proposed GOTLM in (3) is an infinite-dimensional system, where the training ($u_i$) and hyperparameters ($A$ and $b_i$) are all functions in $t$ and $\beta$, respectively, it requires the development of new parameterization, which quantizes the continuous dynamics of both the OT and the ensemble system in (3).

### 3.2.1 ENSEMBLE MOMENTS AND MOMENT SYSTEMS

Here, we develop a parameterization for the GOTLM, by which a finite number of training and hyperparameters are required to represent the OT and the ensemble system dynamics, i.e., to approximate them with quantifiable and controllable precision. Our development is based on the idea of identifying a function with a sequence of numbers (Yosida, 1980). A prominent example is seen

in the domain of probability theory, where one may uniquely determine a distribution (or a Lebesgue density function) by using its moment sequence (See Appendix A). For example, a Gaussian distribution can be determined by its first- (mean) and second-moment (variance). To exploit and attach this property to the GOTLM, we introduce the $k^{\text{th}}$-*ensemble moment* of the system in (3) by

$$m_k(t) = \int_\Omega \beta^k x(t, \beta) d\beta \tag{4}$$

for each $k \in \mathbb{N}$. Then, the state function $x(t, \cdot)$ can be uniquely determined by the associated *moment sequence* $m(t) = \big(m_k(t)\big)_{k \in \mathbb{N}}$ and vice versa (see Appendix A). This identification allows us to derive the governing equation of the moment dynamics, i.e., the moment system, given by the time-derivatives,

$$\frac{d}{dt}m_k(t) = \frac{d}{dt} \int_\Omega \beta^k x(t, \beta) d\beta = \int_\Omega \beta^k \frac{d}{dt} x(t, \beta) d\beta,$$

$$= \int_\Omega \beta^k \big[ A(\beta) x(t, \beta) + \sum_{i=1}^p b_i(\beta) u_i(t) \big] d\beta, \tag{5}$$

where the change of the order of differentiation and integration follows the dominant convergence theorem (Folland, 2013). Because the moment transform $\mathcal{T}$ defined according to (4) by $x(t, \cdot) \mapsto m(t)$ is linear and a vector space isomorphism, the moment system is a linear system controlled by the same inputs $u_i(t)$ defined on the space of moment sequences, of the form

$$\frac{d}{dt}m(t) = \bar{A}m(t) + \sum_{i=1}^p \bar{b}_i u_i(t), \tag{6}$$

where $\bar{A}m(t) = \mathcal{T}_*(Ax_t)$ with $x_t \doteq x(t, \cdot)$ and $\bar{b}_i = \mathcal{T}_* b_i$ are the pushforwards of the vectors fields $Ax_t$ and $b_i$ by $\mathcal{T}$, respectively (see Appendix B). Essentially, $\bar{A}m(t)$ and $\bar{b}_i = \mathcal{T}_* b_i$ are the moment sequences of $Ax_t$ and $b_i$, respectively, as functions on $\Omega$. The constructed moment system is dynamically equivalent to the GOTLM and enables interpretable representation learning of time-dependent OT, which is infeasible by directly using the ensemble system model in (3).

### 3.2.2 OPTIMAL TRANSPORT IN MOMENT REPRESENTATION

As discussed in Section 2, DI characterizes the trajectory of OT from one probability distribution to another. Therefore, an OT task can be realized by perturbing the dynamics of GOTLM to output an ensemble trajectory that tracks the DI trajectory. The learning for controls can be achieved by leveraging the proposed moment representation and moment system. Recall from Section 2 that $\rho_t = (\Phi_t)_{\#}\rho_0$ is the DI trajectory of the transport from $\rho_0$ at $t = 0$ for $t \in [0, 1]$, where $\Phi_t$ denotes the time-dependent transport map, we can then represent the OT trajectory in the moment coordinate, namely,

$$m_k^*(t) = \int_{\mathbb{R}^n} I^k d\rho_t = \int_{\mathbb{R}^n} I^k d(\Phi_t)_{\#}\rho_0 = \int_{\mathbb{R}^n} \Phi_t^k d\rho_0 = \int_{\mathbb{R}^n} [(1-t)I + t\Phi_1]^k d\rho_0,$$

with the dynamics,

$$\frac{d}{dt}m_k^*(t) = \begin{cases} 0, & k = 0, \\ k \int_{\mathbb{R}^n} [(1-t)I + t\Phi_1]^{k-1}(\Phi_1 - I)d\rho_0, & k > 0. \end{cases} \tag{7}$$

We will track the OT moment dynamics in (7) by designing control inputs $u_i(t)$ using the moment system of GOTLM as in (6). In other words, we will learn $u^*(t) = (u_1^*(t), \cdots, u_p^*(t))'$ in (6) such that $m(t) = m^*(t)$ for all $0 \le t \le 1$.

To tackle this tracking problem, we will exploit finite-dimensional truncation of the moment system in (6). Formally, we denote $\widehat{\cdot}$ the truncation operation and let $q$ be the order of truncation. For example, $\widehat{m}^*(t)$ is the truncation of $m^*(t)$ of order $q$, and $\widehat{A}$ is the endomorphism obtained by restricting $\bar{A}$ to the space of truncated moment sequences of order $q$. Algebraically, we may represent any such truncated moment sequence as an $qn \times qn$ matrix, defined by the identity $\widehat{\bar{A}m}(t) = \widehat{A}\widehat{m}(t)$.

Then, the learning problem can be formulated as an optimal tracking problem,

$$K_{pq}: \quad \min_u \int_0^1 \|\widehat{m}^*(t) - \widehat{m}(t)\|^2 dt$$

$$\text{s.t.} \quad \frac{d}{dt}\widehat{m}(t) = \widehat{A}\widehat{m}(t) + \widehat{B}u(t), \quad \widehat{m}^*(0) = \widehat{m}(0), \tag{8}$$

where the truncated OT moment sequences $\widehat{m}^*(t)$ can be computed for a given target distribution, $\|\cdot\|$ is a norm on $\mathbb{R}^{qn}$, $\widehat{B}$ is a $qn \times p$ matrix with columns denoted by $\widehat{b}_i$, $i = 1, \ldots, p$, and $u(t) = (u_1, \ldots, u_p)' \in \mathbb{R}^p$. The goal is to learn the control vector $u(t)$ such that the tracking error $e(t) \doteq \widehat{m}^*(t) - \widehat{m}(t)$ is minimized, where $\widehat{m}^*(t)$ can be viewed as a known reference trajectory to be tracked.

Although this is reduced to a standard optimal tracking problem involving a finite-dimensional linear control system, it is required here to solve a sequence of such problems with respect to different choices of truncation order $q$ and the number of control inputs $p$. We develop a systematic approach by reformulating this learning problem as a sequence of time-varying regression problems. To elaborate, we first observe that the time-evolution of the learning error obeys $\dot{e}(t) = \frac{d}{dt}\widehat{m}^*(t) - \widehat{A}\widehat{m}(t) - \widehat{B}u(t)$ with the initial condition $e(0) = 0$. This then gives, by some algebraic manipulations, the inequality $\|\dot{e}(t)\| \leq \|\frac{d}{dt}\widehat{m}^*(t) - \widehat{A}\widehat{m}(t) - \widehat{B}u(t)\|$. By the Gronwall's inequality, we obtain an upper bound on the learning error, given by $\|e(t)\| \leq \int_0^t \|\frac{d}{ds}\widehat{m}^*(s) - \widehat{A}\widehat{m}(s) - \widehat{B}u(s)\| ds \doteq \alpha(t)$. We know that the convergence of the truncated moment sequences $\widehat{m}^*(t)$ and $\widehat{m}(t)$ to $m^*(t)$ and $m(t)$, respectively, guarantees that $\min \|e(t)\| = \min \alpha(t)$ as the truncation order $p \to \infty$. Therefore, it suffices to learn the control input $u(t)$ minimizing $\|e(t)\|$ at each time $t \in [0, 1]$. This in turn gives rise to a time-varying least-squares problem with the optimal solution given by

$$u^*(t) = (\widehat{B}'\widehat{B})^{-1}\widehat{B}'\Big[\frac{d}{ds}\widehat{m}^*(t) - \widehat{A}\widehat{m}(t)\Big], \tag{9}$$

provided that $\widehat{B}$ is of full-column rank $p$; or equivalently, the columns of $\widehat{B}$, $\widehat{b}_1, \ldots, \widehat{b}_p$ are linearly independent over $\mathbb{R}^{qn}$. Condequently, the GOTLM driven by the learned optimal control $u^*(t)$, i.e., $\frac{d}{dt}x(t, \beta) = f(\beta)x(t, \beta) + \sum_{i=1}^p b_i(\beta)u_i^*(t)$, gives the desired ensemble system representation of the OT from $\rho_0$ to $\rho_1$. This learning algorithm is displayed in Algorithm 1.

---

**Algorithm 1** Generalized OT-learning model

---

**Input:** $\rho_0$, $\rho_1$, $A$, and $b_i$ for $i = 1, \ldots, q$
**Output:** $x^*(t, \beta)$
  *Initialization* : Given $p$ and $0 = t_0 \leq t_1 \leq \cdots \leq t_M = 1$
2: Compute the OT map $\Phi$.
  Find the order $p$ truncated moment parameterization $\widehat{A}$, $\widehat{b}_i$ for $i = 1, \ldots, p$, and $\widehat{m}^*(t)$ for $0 \leq t \leq 1$
4: Solve $v(\widehat{m}(t)) = \text{argmin}_{a \in \mathbb{R}^{qn}} \|\frac{d}{dt}\widehat{m}^*(t) - \widehat{A}\widehat{m}(t) - \widehat{B}a\| = (\widehat{B}'\widehat{B})^{-1}\widehat{B}'\big(\frac{d}{dt}m^*(t) - \widehat{A}\widehat{m}(t)\big)$
  in terms of $\widehat{m}(t)$ for all $0 \leq t \leq 1$, where $\widehat{B} = [\,\widehat{b}_1 \mid \cdots \mid \widehat{b}_q\,]$
  Solve $\frac{d}{dt}\widehat{m}(t) = \widehat{A}\widehat{m}(t) + \widehat{B}v(\widehat{m}(t))$ by an ordinary differential equation solver for $\widehat{m}(t)$ on $t \in [0, 1]$ with the initial condition $\widehat{m}(0) = \widehat{m}^*(0)$
6: $u^*(t) \leftarrow v(\widehat{m}(t))$ as a function of $t \in [0, 1]$
  Compute $x^*(t, \beta) = e^{tA(\beta)}x_0(\beta) + \int_0^t e^{(t-s)A(\beta)}B(\beta)u^*(t)dt$ with $x_0$ the density function of $\rho_0$ and $B(\beta) = [\,b_1(\beta) \cdots b_q(\beta)\,]$
8: **return** $x^*(t, \beta)$

---

## 4  EXAMPLES AND NUMERICAL SIMULATIONS

In this section, the applicability of Algorithm 1 will be demonstrated through two illustrative examples, including OT between Lebesgue density functions and probability distributions over a compact and a non-compact support along the geodesics. In particular, we will show that in these two different cases, the trajectories of the learned GOTLMs converge to the OT trajectories in different modes.

## 4.1 Optimal transport of Distributions over a compact support

To illustrate the idea and operation of our dynamical systems modeling and learning approaches for OT, without loss of generality, we consider probability distributions with the support $\Omega = [0, 1]$. Our first illustration is to consider OT between two probability density functions.

**Example 1** In this example, we consider OT from a square to a triangle wave representing probability density functions $x_0(\beta) = 1$, $\beta \in [0, 1]$, and $x_1(\beta) = \frac{8}{3}\beta\mathbb{1}_{[0,\frac{3}{4}]}(\beta) - 8(\beta - 1)\mathbb{1}_{(\frac{3}{4},1]}(\beta)$, respectively, in which $\mathbb{1}_I$ is the indicator function of the set $I \subset \Omega$, that is, $\mathbb{1}_I(\beta) = 1$, if $\beta \in I$, and $\mathbb{1}_I(\beta) = 0$, otherwise. Let $X_i$ denote the cumulative distribution function of $x_i$ for $i = 0, 1$, then the transport map is given by $\Phi_1(\beta) = X_1^{-1} \circ X_0(\beta) = \frac{\sqrt{3\beta}}{2}\mathbb{1}_{[0,\frac{3}{4}]}(\beta) + \left(1 - \frac{\sqrt{1-\beta}}{2}\right)\mathbb{1}_{(\frac{3}{4},1]}(\beta)$. This yields the moment parameterized DI dynamics as $\frac{d}{dt}m_0^*(t) = 0$ and $\frac{d}{dt}m_k^*(t) = \int_0^1 k\big((1 - t)\beta + t\Phi_1(\beta)\big)^{k-1}(\Phi_1(\beta) - \beta)x_0(\beta)d\beta$ for $k \geq 1$. For example, the dynamics of the first three orders are $\frac{d}{dt}m_1^*(t) = \frac{1}{12}$, $\frac{d}{dt}m_2^*(t) = \frac{7t}{480} + \frac{3}{160}$, and $\frac{d}{dt}m_3^*(t) = \frac{t^2}{448} + \frac{19t}{2240} + \frac{19}{6720}$.

Now, let us consider a GOTLM of the form,

$$\frac{d}{dt}x(t, \beta) = \beta x(t, \beta) + \sum_{i=1}^{p} \beta^{i-1}u_i(t), \tag{10}$$

where $A(\beta) = \beta$, and $b_i(\beta) = \beta^{i-1}$, $i = 1, \dots p$, and $\beta \in [0, 1]$. Then, the associated moment system cay be derived using the transform in (4), which gives

$$\frac{d}{dt}m_k(t) = m_{k+1}(t) + \sum_{i=1}^{p} \frac{1}{k+i}u_i(t), \quad k \in \mathbb{N}. \tag{11}$$

We then construct problem $K_{pq}$ as presented in (8), where the truncated moment system of order $q$ driven by $p$ control inputs takes the form,

$$\frac{d}{dt}\widehat{m}(t) = \widehat{A}\widehat{m}(t) + \widehat{B}u(t)$$

$$= \begin{bmatrix} 0 & 1 & & \\ & 0 & \ddots & \\ & & \ddots & 1 \\ & & & 0 \end{bmatrix} \begin{bmatrix} \widehat{m}_0(t) \\ \widehat{m}_1(t) \\ \vdots \\ \widehat{m}_q(t) \end{bmatrix} + \begin{bmatrix} 1 & \frac{1}{2} & \cdots & \frac{1}{p} \\ \frac{1}{2} & \frac{1}{3} & \cdots & \frac{1}{p+1} \\ \vdots & \vdots & \ddots & \vdots \\ \frac{1}{q+1} & \frac{1}{q+2} & \cdots & \frac{1}{q+p} \end{bmatrix} \begin{bmatrix} u_1(t) \\ u_2(t) \\ \vdots \\ u_p(t) \end{bmatrix}. \tag{12}$$

where $\widehat{m}(t) \in \mathbb{R}^{q+1}$, $\widehat{A} \in \mathbb{R}^{(q+1)\times p}$, $\widehat{B} \in \mathbb{R}^{(q+1)\times p}$, and $u_i \in \mathbb{R}^p$. We applied Algorithm 1 to solve this problem, where we set $p = 7$ and $q = 10$. The learned control inputs are shown in Figure 2b. As the model confines the evolution time in $[0, 1]$, the input controls have overshoots initially. This effect can be diluted by re-scaling the time period, e.g., simply to $[0, T]$ for $T > 1$. The time-evolution of the ensemble states in (10) following the learned control inputs are shown in Figure 2d, which are approaching the desired transport interpreted by DI, as shown in Figure 2c, as $t \to 1$.

The performance of the constructed GOTLM can be made better by increasing the values of $(p, q)$, i.e., by adding more control inputs and enhancing the approximation precision of the moment system in (10) to the GOTLM in (11). In this example, we evaluation such convergence behavior by varying $p$ from 1 to 10 and $q$ from 2 to 10, under the condition $p < q$ to formulate well-defined least-squares problems of $K_{pq}$. Figure 3 shows the sum of squares error between the final and the target state for both the ensemble and truncated moment systems, i.e., $\int_0^1 |\widehat{x}_1(\beta) - x_1(\beta)|^2 d\beta$ and $\sum_{k=0}^{100} |\widehat{m}_k(1) - m_k^*(1)|$, with respect to $(p, q)$, where $\widehat{x}_1$ denotes the final state of the controlled ensemble system in (10), and $\widehat{m}_j = 0$, for $j > q$. This indicates the convergence with respect to $(p, q)$, specifically in the $L^2$ and $\ell^2$ sense for the ensemble and the moment system, respectively. Theoretically, the learned trajectory will converge to the desired DI trajectory as $p, q \to \infty$.

## 4.2 Optimal transport of Distributions over a non-compact support

In this section, we illustrate the application of our method for OT of distributions defined on a non-compact support.

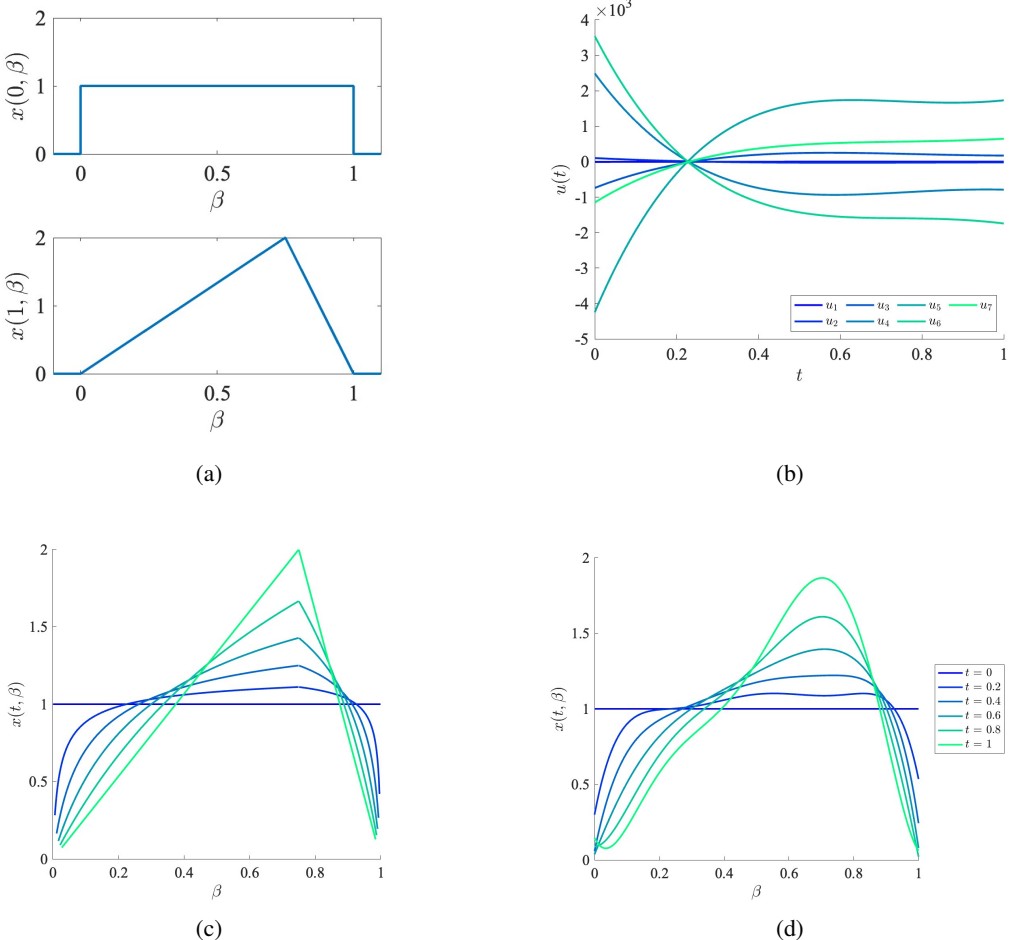

Figure 2: OT of compactly supported distributions for $p = 7$ and $q = 10$. (a) Initial (top) and final distribution (bottom); (b) learned control inputs; (c) DI trajectories; and (d) ensemble system trajectories.

**Example 2** We use the example of well-studied OT between one-dimensional normal distributions to demonstrate our learning representation approach. Because normal distributions are defined over $\mathbb{R}$, we introduce a discount factor in the GOTLM to guarantee that the learned ensemble system remains bounded as $\beta \to \infty$. In particular, we consider the GOTLM of the form,

$$\frac{d}{dt}x(t,\beta) = \beta x(t,\beta) + \sum_{i=1}^{p} \frac{e^{-|\beta|}}{(i-1)!} \beta^{i-1} u_i(t),\tag{13}$$

where $A(\beta) = \beta$, $b_i(\beta) = \frac{\beta^{i-1}}{(i-1)!}$, $i = 1, \ldots, p$, and $e^{-|\beta|}$ is a discount function compensating for the growth of the state $x(t,\beta)$ with respect to $\beta$. In addition, because the statistical moments of normal distributions form an increasing sequence, we introduce a discount factor to the ensemble moments of the system in (13) as well, by a rescaling, i.e., $m_k(t) = \frac{1}{k!} \int_{-\infty}^{\infty} \beta^k x(t,\beta) d\beta$, which leads to the moment system,

$$\frac{d}{dt}m_k(t) = (k+1)m_{k+1}(t) + \sum_{i=1}^{m} \binom{k+i}{i} [1 + (-1)^i] u_i(t).\tag{14}$$

To fix ideas, we consider the OT from the normal distribution $\mathcal{N}(0,1)$ to $\mathcal{N}(1,1)$, where we chose $p = 10$ and $q = 10$. The learned controls $u_i(t)$ are plotted in Figure 4b, and the the final state the ensemble system in (13) is shown in Figure (4a), which agrees with the desired transport to $\mathcal{N}(1,1)$.

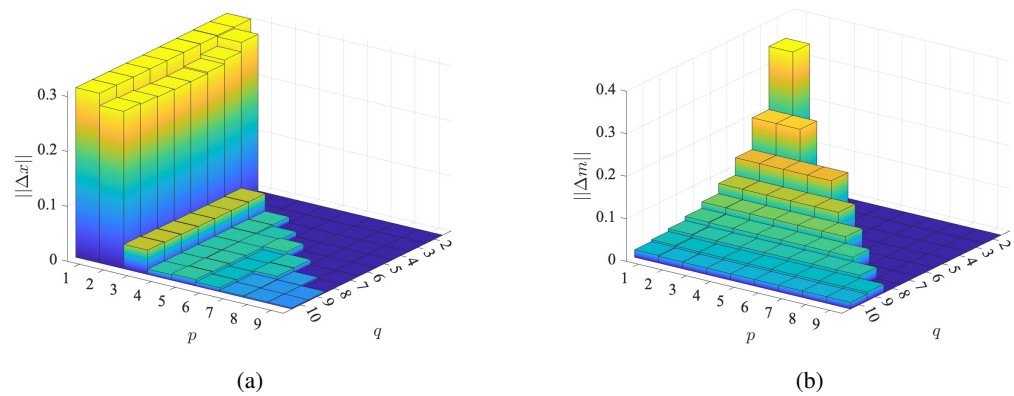

Figure 3: Sum of squares error with respect to the number of control $p$ and order of truncation $q$ between the final and the target (a) states and (b) moment sequences.

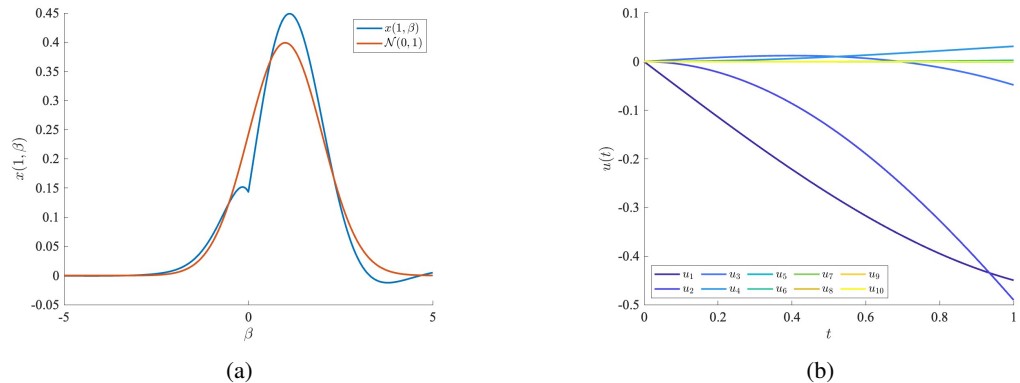

Figure 4: OT of normal distribution transport from $\mathcal{N}(0,0)$ to $\mathcal{N}(1,0)$ for $p = 10 \, q = 10$ . (a) Final state distribution, and (b) control inputs.

**Example 3** In practice, to avoid evaluating distributions defined on unbounded domains, it is common to approximate such distributions by those with compact supports, e.g., the use of confidence intervals in statistical learning theory (Cumming & Calin-Jageman, 2017). To incorporate this idea into the proposed GOPLM framework, we consider a relaxed transport from $\mathcal{N}(0, 0.5^2)$ to $\mathcal{N}(0.2, 0.3^2)$ with desired confidence, quantified by a confidence interval. Specifically, in this example, we require more than $95\%$ of mass concentrated on the compact interval $\Omega = [-1, 1]$ after the transport. Denoting their probability density functions by $p_0$ and $p_1$, respectively, we aim to learn the representation of the transport from $x_0 = p_0 \mathbb{1}_{[-1,1]}/\rho_0(\Omega)$ to $x_1 = p_1 \mathbb{1}_{[-1,1]}/\rho_1(\Omega)$, where $\rho_0(\Omega)$ and $\rho_1(\Omega)$ are normalization constants for $x_0$ and $x_1$ to be probability density functions. Due to the compactness of $\Omega$, it is possible to use GOPLM presented in (10) to represent this transport. Here, we choose $p = 10$ and $q = 10$. The time-evolution of the ensemble states and the final state steered by the learned control inputs are shown in Figures **??** and **??**, where we observe that the desired transport is completed.

## 5  CONCLUSION

In this work, we proposed a generalized OT-learning model (GOTLM) to learn an ensemble system representation of OT. The central idea is to use the controls applied to the system as the learnable parameters to train GOTLM, so that GOTLM outputs the DI trajectory connecting the desired source and target probability distributions. In particular, for the purpose of effective model training, we developed the moment representation to parameterize the model, and hence the ensemble system, in

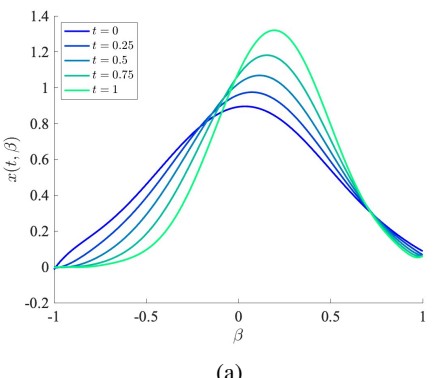
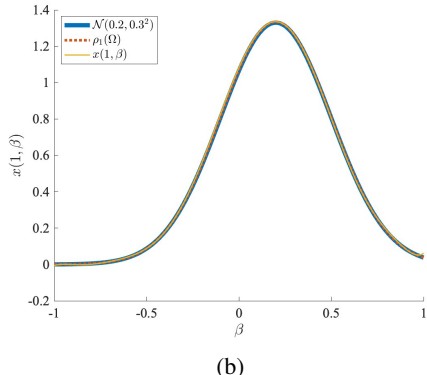

(a)                 (b)

Figure 5: OT of concentrated normal distribution transport from $\mathcal{N}(0, 0.5^2)$ to $\mathcal{N}(0.2, 0.3^2)$ for $p = 10$, $q = 10$. (a) System trajectories of the ensemble system, and(b) final state distribution.

terms of moment sequences. In turn, GOTLM further gives rise to a systematic optimal transport-based approach to learning optimal controls for ensemble systems. The applicability of GOTLM is then demonstrated by constructing ensemble system representations of OT between probability distributions with both compact and non-compact supports. Moreover, it is worth noting that the use of moments also indicates the applicability of GOTLM in a purely data-driven environment, which sheds light on broadening the scope of GOTLM to include tasks, such as data-driven control, pattern recognition, and image classification, to its application domain.

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
