# A  THE METHOD OF MOMENTS

The method of moments was developed by the Russian mathematician Pafnuty L. Chebyshev in 1887 for proving the central limit theorem, and then rigorously formulated by his student Andrey A. Markov (Fischer, 2010). The main idea is to represent probability distributions in terms of infinite sequences of real numbers, because functional representations, e.g., probability density functions, cumulative distribution functions, and characteristic functions, of many probability distributions in practice, may be intractable. Specifically, given a probability measure $\mu$ on the real line, then the $k^{\text{th}}$ moment is defined by

$$m_k = \int_{-\infty}^{\infty} x^k d\mu(x)$$

for each $k \in \mathbb{N}$ A natural question is whether the probability measure $\mu$ can be uniquely determined by the sequence of moments $\{m_k\}_{k \in \mathbb{N}}$ and vice vera.

**Theorem 1** *Let $\mu$ be a probability measure on the real line having finite moments $m_k$ of all orders. If the power series $\sum_{k=1}^{\infty} m_k r^k / k!$ has a positive radius of convergence, then $\mu$ is the only probability measure with the moments $m_k$ for $k \in \mathbb{N}$*

*Proof.* See (Billingsley, 1995). □

Of course, if $\mu$ is compactly supported, the condition in Theorem 1 is satisfied. To see this, we suppose the support of $\mu$ is contained in the compact interval $[a, b]$, then $m_k = \int_{-\infty}^{\infty} x^k d\mu(x) = \int_a^b x^k d\mu(x) \leq M^k$ holds for each $k$ since $\mu([a, b]) = 1$, where $M = \max\{|a|, |b|\}$. As a result, we have $\sum_{k=0}^{\infty} m_k r^k / k! \leq \sum_{k=0}^{\infty} (Mr)^k / k! = e^{Mr}$, which has the infinite radius of convergence. Typical moment-determined probability distributions with non-compact supports are normal distributions. For example, the normal distribution $\mathcal{N}(0, \sigma)$ has the moments $m_k = \sigma^k (k-1)!! = \sigma^k (k-1)(k-3) \cdots 3 \cdot 1$ for $k$ even and $m_k = 0$ otherwise, and then we have $\sum_{k=0}^{\infty} m_k r^k / k! = \sum_{k=0}^{\infty} (\sigma r)^{2k} / (2k)!! = \sum_{k=0}^{\infty} (\sigma r)^{2k} / 2^k k! = e^{(\sigma r)^2 / 2}$, which also has the infinite radius of convergence.

In the early 20 century, the notion of moments was extended from probability measures to general Borel measures, and the problem of the existence and uniqueness of a moment sequence representing a Borel measure is referred to as the moment problem, which has already been extensively studied under different settings. Those relevant to our work are the Hausdorff moment problem and Hamburger moment problem, which concern with Borel measures supported on $[0, 1]$ and the entire $\mathbb{R}$, respectively.

**Theorem 2** *Let $\{m_k\}_{k=0}^{\infty}$ be a sequence of real numbers.*

1. *(Hausdorff moment problem) There is a unique Borel measure on $[0, 1]$ such that $m_k = \int_0^1 x^k d\mu(x)$ for all $k \in \mathbb{N}$ if and only if $\sum_{i=0}^n \binom{n}{i} (-1)^i m_{k+i} \geq 0$ for all $k, n \in \mathbb{N}$.*

2. *(Hamburger moment problem) There is a unique Borel measure on $\mathbb{R}$ such that $m_k = \int_{-\infty}^{\infty} x^k d\mu(x)$ for all $k \in \mathbb{N}$ if and only if $\sum_{i,j \in \mathbb{N}} m_{j+k} c_i \bar{c}_j \geq 0$ for any sequence $\{c_k\}_{k \in \mathbb{N}}$ of complex numbers such that $c_k = 0$ holds for all but finitely many $k$, equivalently, the Hankel matrix*

$$H_n = \begin{bmatrix} m_0 & m_1 & \cdots & m_n \\ m_1 & m_2 & \cdots & m_{n+1} \\ \vdots & \vdots & & \vdots \\ m_n & m_{n+1} & \cdots & m_{2n} \end{bmatrix}$$

*has positive determinant for all $n \in \mathbb{N}$.*

*Proof.* See (Hausdorff, 1923; Hamburger, 1920). □

# B  DIFFERENTIAL CALCULUS IN MOMENT COORDINATES

In this work, we are particularly interested in Borel measures on $\Omega \subseteq \mathbb{R}^n$ that are absolutely continuous with respect to the Lebesgue measure with the Radon-Nikdym derivatives given by $\mathbb{R}^n$-valued

$L^p$ functions for some $p \geq 1$. Let $\mathcal{M}$ denote the space of moment sequences associated with $L^p$-functions, then the moment problem implies that the moment transform $\mathcal{T} : L^p(\Omega, \mathbb{R}) \to \mathcal{M}$ is a bijective map. It is straightforward to observe that $\mathcal{M}$ is a vector space over $\mathbb{R}$: if $m$ and $m'$ are the moment sequences associated with $f$ and $f'$, then $am + a'm'$ is necessarily the moment sequence associated with $af + a'f'$ for any $a, a' \in \mathbb{R}$. This further proves that $\mathcal{T}$ is a linear map, and hence a vector space isomorphism.

Now, we equip the space of moment sequences $\mathcal{M}$ the quotient topology generated by $\mathcal{T}$ so that $\mathcal{T}$ becomes a continuous linear map, then $\mathcal{T}$ also becomes differentiable (actually smooth) with the differential $D\mathcal{T}$ being able to be identified with $\mathcal{T}$ as well. This provides the tool for calculating the moment representation of a vector field on $L^p$ in terms of its pushforward by $\mathcal{T}$. For example, given a linear vector field $V$ on $L^p(\Omega, \mathbb{R}^n)$, then there is an $\mathbb{R}^{n \times n}$ valued function $A$ defined on $\Omega$ such that $V = Af \in L^p(\Omega, \mathbb{R}^n)$ for any $f \in L^p(\Omega, \mathbb{R}^n)$. Then, by the definition of pushforward vector fields, the pushforward of $V$ is a vector field on $\mathcal{M}$, given by $\mathcal{L}_* V = D\mathcal{T} \cdot A \cdot \mathcal{T}^{-1} m = \mathcal{T} \cdot A \cdot \mathcal{T}^{-1} m$ with $m$ denoting the moment sequence of $f$, which is again a linear vector field because the composition $\mathcal{T} \cdot A \cdot \mathcal{T}^{-1}$ of linear maps is still linear.