# OpenReview forum: "Dynamic Representation of Optimal Transport via Ensemble Systems"
_ICLR.cc/2024/Conference — Submitted to ICLR 2024_

### Official Review · Reviewer_wGDR · 2023-10-23

**Soundness:** 4 excellent
**Presentation:** 3 good
**Contribution:** 1 poor
**Rating:** 3
**Confidence:** 4

**Summary:**

Isabel Haasler et al. recently considered the connections of Control and Estimation of Ensembles and Structured Optimal Transport (see, e.g., their survey in IEEE CSM https://ieeexplore.ieee.org/document/9491021). The authors elaborate upon these connections.

**Strengths:**

The topic is interesting and the paper is mathematically rigorous.

The examples of Section 4 are well chosen and illustrate the problem appropriately.

**Weaknesses:**

It is not clear what is the actual contribution. Much of the text, up to and including Eq. 8 essentially summarizes the material in IEEE CSM (https://ieeexplore.ieee.org/document/9491021). After Eq. 9 starts the "Examples and Numerical Simulations" section and there are no theorems in the supplementary material either.

There are some missing Figures / wrong labels on page 8 (two Figures referenced as ?? and ??).

**Questions:**

Please read the review in IEEE CSM (https://ieeexplore.ieee.org/document/9491021) and explain clearly what is the actual contribution.

---

### Official Review · Reviewer_Qcm9 · 2023-10-25

**Soundness:** 2 fair
**Presentation:** 2 fair
**Contribution:** 2 fair
**Rating:** 5
**Confidence:** 4

**Summary:**

In the submitted manuscript, the authors present a novel approach to constructing a machine learning model for optimal transport processes. They formulate the model as an ensemble control system defined over a function space and employ moment representation techniques. Furthermore, the authors develop a methodology for computing the optimal control law. Below is a summary of the paper's strengths and weaknesses.

**Strengths:**

1. **Concrete Derivation**: I commend the authors for their rigorous work in utilizing moment representation techniques to model moment dynamics, as well as for their solution to the optimal tracking problem.
2. **Highly Related Topic**: The topic of this paper is highly related to the conference.

**Weaknesses:**

1. **Conservation law** As is widely accepted in the field, the physical interpretation of transport processes fundamentally relies on the principles of fluid mechanics. In this context, equations describing such processes should adhere to the Fokker-Planck equation, $\frac{\partial \rho}{\partial t} = -\nabla \cdot (v\rho)$, where $\rho$ represents the probability density function and $v$ denotes the velocity field. Notably, Equation (3) in the submitted manuscript diverges from this established theoretical framework. According to Reference [1], the phenomenon under discussion would be more accurately categorized as 'teleportation' rather than 'transportation'.
2. **Solving optimal control problem** I observe that the system under consideration is time-invariant. Given this characteristic, I find it perplexing that the authors have chosen to design the control law based on the 'separation principle'—unless I am mistaken in my understanding. Furthermore, I would argue that Algorithm 1 appears to be superfluous. A simpler formulation could be derived for this optimal control problem, which essentially constitutes a 'two-point boundary value problem.' This simplification is possible because the state $\hat{m}$ can be directly obtained.
3. **The end point time** Why have the authors chosen to set the endpoint time at $t = 1$? This particular choice could be better justified, potentially by referencing related work in the field of Schrödinger bridge theory

4. **Proof of controllability/reachability** The manuscript lacks a proof of controllability/reachability for Equation (8). In other words, it remains unclear whether the proposed Algorithm 1 can definitively solve Equation (8).

5. **Complexity**  In my opinion, the manuscript would be strengthened by the inclusion of computational time and space complexity metrics. Given that the ODE solver employed likely necessitates discretization, there are potential implications for both space and time complexity that could be significant. Therefore, I recommend that the authors provide a detailed analysis of the computational complexities involved.

6. **Baseline Comparison**  The authors should consider incorporating baselines that account for both optimal control solvers and filtering issues in the experiments. According to my understanding, the optimal control problem can be converted into a nonlinear programming problem or filtering problem. The nonlinear programming problem can subsequently solved using solvers like IpOpt [2], AntiGONE, as detailed on page 35 in Reference [3]. Concurrently, this problem can also be addressed through filtering techniques (duality of control and filtering).
-----
Reference :
[1]. Neklyudov et al. Wasserstein Quantum Monte Carlo: A Novel Approach for Solving the Quantum Many-Body Schrödinger Equation
[2]. Biegler L T, Zavala V M. Large-scale nonlinear programming using IPOPT: An integrating framework for enterprise-wide dynamic optimization[J]. Computers & Chemical Engineering, 2009, 33(3): 575-582.
[3]. Biegler numero.cheme.cmu.edu/content/06720/Dynopt2.pdf

**Questions:**

See Weakness.

---

### Official Review · Reviewer_vm8m · 2023-11-01

**Soundness:** 2 fair
**Presentation:** 1 poor
**Contribution:** 2 fair
**Rating:** 3
**Confidence:** 2

**Summary:**

The paper introduces a novel perspective on optimal transport (OT) and its potential applications in dynamical systems and control. The authors propose an ensemble-systems framework to model the optimal transport process as a dynamic system. They achieve this by utilizing moment kernel representations to describe the dynamics of optimal transport and ensemble systems.

One of the key contributions of this work is the development of a generalized OT-learning model (GOTLM) that learns an ensemble system representation of OT. In this model, the controls applied to the system are used as learnable parameters to enable the generation of displacement interpolation trajectories between source and target probability distributions.

**Strengths:**

The major contribution of this paper is to merge the domains of optimal transport (OT) and dynamic systems control. It introduces an innovative approach by establishing an ensemble-systems interpretation for modeling the optimal transport process. Specifically, the utilization of moment kernel representations to describe the dynamics of optimal transport and ensemble systems is a novel and distinctive aspect of this work.

The paper exhibits significant jumps and discontinuities in its presentation, making it challenging for readers to follow and understand the presented concepts.  (See below.) It is difficult to assess the overall quality and significance of the work.

**Weaknesses:**

1. The foundation of this paper, expressed through Equation (3), is essential for understanding the transformation of the OT problem into a control problem. However, the paper fails to provide any explanation or proof for why the OT problem can be formulated in the form of Equation (3). The relationships between the variables, including x, A, b, beta, and u, and the original OT problem remain unclear. The origins of the dimensions n and p are not adequately justified. To enhance the paper, a comprehensive theoretical exposition of Equation (3) and its connections to the core OT problem is needed.

2. In Equation (4), there's an integral related to beta. It's not clear whether this integral always exists. Commonly, when calculating moments, integrals are associated with specific probability measures, but in this integral, no probability distribution is specified.

3. In Equation (7), I and I^k are introduced, but the meaning and relevance of these variables are unclear. Does "I" still represent the identity map here?

4. Equation (8) mentions that $\hat m^*(t)$ can be computed for a given target distribution, but the specific methodology for computing it is not provided.

5. The derivations from Equation (3) to Equation (8) are highly intricate and challenging to follow. The paper should consider providing more detailed explanations, even if the specific proofs are placed in an appendix. The focus should shift toward making the theoretical framework more accessible, with clear and intuitive explanations.

6. It would be valuable for the paper to explore whether the moment system methodology can be extended to use more general basis functions, including kernel functions.

7. In optimal control problems, it is common to consider the norm of control inputs u as an optimization objective to prevent excessive inputs. The paper lacks discussion or consideration of such an energy constraint in the optimization process.

8. The examples provided in the paper are overly simplistic, limited to one-dimensional cases, and lack comparison with existing methods. This simplicity hinders the ability to evaluate the significance of the paper's contributions.

**Questions:**

See the above

---

### Official Review · Reviewer_DFD4 · 2023-11-02

**Soundness:** 2 fair
**Presentation:** 2 fair
**Contribution:** 2 fair
**Rating:** 5
**Confidence:** 3

**Summary:**

The paper explores the dynamic formation of optimal transport through the use of displacement interpolation and then proposes a dynamical system, called ensemble process, whose purpose would be to track the moments (a number of them) from the trajectory described by the DI. An Algorithm is proposed, which chooses the right control signals to be able to track such moments. Examples and numerical simulations are given.

**Strengths:**

-> The paper makes a good connection between tools used in control and systems to solve a problem, namely, approximating the moments of a dynamic system described by optimal transport.

-> An algorithm is provided, and its derivation is shown at great extent.

**Weaknesses:**

I have many concerns regarding the paper.

-> I fail to understand the real contribution of the paper, or at least see if it matches what I understood the paper believes to be its contribution. The paper claims to provide an “ensemble-systems interpretation for modeling the optimal transport process”, and I would expect that such interpretation based on dynamical systems would somehow contribute to the understanding of optimal transport, perhaps from a theoretical perspective. But the paper, instead, proposes a dynamical system driven by control signals that the paper will use to track the dynamics of the moments of an optimal transport (OT) path according to displacement interpolation (DI). The problem with this is that there is nothing particularly “special” about the dynamics the authors have “proposed” because, in principle, one could use other dynamical systems with control inputs to try to solve the same tracking problem. I don’t see how the dynamical systems that they propose help “interpreting” the modeling the optimal transport processes.

Moreover, the paper claims in its contribution that “based on the time-dependent description of OT in terms of displacement interpolation (DI), we construct a machine learning model that represents an OT process in the form of an ensemble control system defined on a function space. In this model, the control inputs are learnable parameters used to track the OT dynamics.” Well, there is no such “machine learning model”: as I mentioned before, the system proposed by the authors is just a dynamical system that is used as a tool to track the dynamics of optimal transport. I don’t see how this is a machine learning model. I find the contribution statement by the authors misleading.

Also, if what truly the authors are doing is using a machine learning model to solve a control problem, where is such model? I don’t see why Algorithm 1 would be considered an ML algorithm, looks more like a control or estimation algorithm.

-> The paper says the following in the conclusion: “Moreover, it is worth noting that the use
of moments also indicates the applicability of GOTLM in a purely data-driven environment, which
sheds light on broadening the scope of GOTLM to include tasks, such as data-driven control, pattern
recognition, and image classification, to its application domain.” The problem with such statement is that the paper does not present any result regarding such important applications! If I were to see data-driven control, pattern recognition, and image classification tasks being solved using GOTLM, then I would feel the paper is closer to be an ML paper because I would take Algorithm 1 as a numerical solution to be employed in ML tasks. However, the paper is missing this great opportunity and I believe makes it less relevant!

-> Finally, there is something that makes no sense in Algorithm 1. Why does it need to compute the OT map $\phi$? That in itself seems to be a very hard problem (how to solve this in a closed-form manner?), and seems to contradict the practical use of GOTLM, which is to track the moments of the OT dynamics without even knowing the OT map! This decreases the contribution of the paper, since Algorithm 1 becomes impractical. Please, clarify.

==

-> Equation (3) is introduced as “ generalized OT-learning model (GOTLM)”, and two paragraphs after the equation it is mentioned, regarding the same equation, that “This learning representation model maps the input $x_0$ to the output $x_T$ tracking the OT trajectory by tuning the control inputs.” However, up to this point there is nothing truly related to “optimal transport” since there is no optimization that one is trying to solve and specify the controls for: we just have a dynamical system like the one can find on a control a systems book. It is not casted as an optimal control problem at that point of the paper.

-> Also, when introducing the states of the ensemble system, there is no interpretation on what the curve $X(t,\dot)$ represents. It is a little bit later that one has to guess that maybe it perhaps represents a probability density function. What is it? Now, bear in mind that displacement interpolation (ID) can be specified for measures that are not necessarily absolutely continuous with respect to the Lebesgue measure, and so do not need to have a density function necessarily! The paper must specify the class of measures it is analyzing.

-> $\beta$ is called a parameter when being introduced in equation (3), but what does it really represent in terms of probability density functions? Given that $x(t,\dot)\in F(\Omega,R^n)$ right after equation (3), makes me think it is the domain over which the density is defined on, but this wouldn’t make sense since $\Omega$ is a subset of the real line, whereas Figure 1 indicates that the domain can have more dimensions. Please clarify.

-> Something I don’t understand is: how many moments do we need to do the right approximation of the ensemble to the OT problem? Nothing of this is explained.

==

-> In section 3.2.1, I am confused about the use of notation. The operator $\mathcal{T}$ is defined to take $x(t,\dot)$ and make it a moment. However, of which order? What order of moment does $\mathcal{T}_{*}$ represent? This is not clear to me and makes reading the whole section confusing and hard to understand.

-> The symbol $m_k$ is used to represent the $k$ th moment; however, the symbol $m$ is introduced even before it is properly defined as a vector of truncated moments until moment $q$. for example $m$ is used in the paragraph after equation (7) without definition. Make sure the paper only refers to the right moments throughout. Please, emphasize the dimension of $m$.

-> What is $I^k$ in the unnumbered equation before equation (7)? It does not seem to be the identity operator, otherwise, we would have $I^k = I$ which wouldn’t give us $\Phi^k_t$.

-> Right before equation (9) it says “truncation order $p\to\infty$, but this is confusing, since the truncation error seems to be $q$. Please, clarify.

-> What is the $f(\beta)$ in the paragraph before Algorithm 1?

==

-> Is the term “ensemble system” standard in the literature of dynamical systems, especially for linear systems or perhaps affine-control systems? The term is mentioned after equation (3) without much further explanation.

-> Define notation of function spaces to make it more self-contained, e.g., what are $L_p$ spaces, how the domain and range are defined in the notation, etc.

-> “Consequently” instead of “Condequently” after equation (9).

-> The literature review may need more revision of works and a more thorough presentation of applications of displacement interpolation. For example, googling "displacement interpolation Neurips", led me to the paper "Dynamical Wasserstein Barycenters for Time-series Modeling" by Kevin Cheng, Shuchin Aeron, Michael C. Hughes, Eric L Miller. A google search on "displacement interpolation IEEE" led me to the paper "Distributed Wasserstein Barycenters via Displacement Interpolation" by Pedro Cisneros-Velarde, Francesco Bullo. Both papers use the concept of displacement interpolation: the former for time-series modeling, and the second for consensus-based algorithms.

**Questions:**

Please, see the "Weaknesses" section.

---

### Meta-Review · Area_Chair_kPSF · 2023-12-05

**Metareview:**

The reviewers all agreed that there were important problems with this submission, and the authors did not reply.

**Justification For Why Not Higher Score:**

There was no reply from the authors addressing the reviewers' comments.

**Justification For Why Not Lower Score:**

reject

---

### Decision · Program_Chairs · 2024-01-16

Reject